# Factors and consequences associated with intimate partner violence against women in low- and middle-income countries: A systematic review

Lakma Gunarathne[1]*, Jahar Bhowmik[1], Pragalathan Apputhurai[1], Maja Nedeljkovic[2]

**1** Department of Health Science and Biostatistics, Swinburne University of Technology, Hawthorn, Victoria, Australia, **2** Department of Psychological Sciences, Swinburne University of Technology, Hawthorn, Victoria, Australia

* lgunarathne@swin.edu.au

**Data Availability Statement:** All relevant data are within the manuscript and its Supporting Information files.

## Abstract

Intimate Partner Violence (IPV) is a global public health issue, with notably high prevalence rates observed within Low-and Middle-Income Countries (LMICs). This systematic review aimed to examine the risk factors and consequences associated with IPV against women in LMICs. Following PRISMA guidelines, we conducted a systematic review using three databases: Web of Science, ProQuest Central, and Scopus, covering the period from January 2010 to January 2022. The study included only peer-reviewed journal articles in English that investigated IPV against women in LMICs. Out of 167 articles screened, 30 met the inclusion criteria, comprising both quantitative and mixed-method studies. Risk factors of IPV were categorised as: demographic risk factors (23 studies), family risk factors (9 studies), community-level factors (1 studies), and behavioural risk factors (14 studies), while consequences of IPV were categorised as mental health impacts (13 studies), physical impacts (5 studies), and societal impacts (4 studies). In this study, several risk factors were identified including lower levels of education, marriage at a young age, poor wealth indices, rural residential areas, and acceptance of gender norms that contribute to the prevalence of IPV in LMICs. It is essential to address these factors through effective preventive policies and programs. Moreover, this review highlights the necessity of large-scale, high-quality policy-driven research to further examine risk factors and consequences, ultimately guiding the development of interventions aimed at preventing IPV against women in LMICs.

## Introduction

Intimate Partner Violence (IPV) is a significant global public health issue [1]. According to World Health Organization (WHO), IPV is defined as behaviour by an intimate partner or ex-partner that causes physical, sexual, or psychological harm, including physical aggression, sexual coercion, psychological abuse, and controlling behaviours [1]. The WHO reports that one out of three women aged 15 to 49 have been exposed to IPV [2]. Therefore, women are at a

**Funding:** This study was financially supported by Swinburne University of Technology in the form of a PhD study scholarship awarded to LG. No additional external funding was received for this study. The funder had no role in study design, data collection and analysis, decision to publish, or preparation of the manuscript.

**Competing interests:** The authors have declared that no competing interests exist.

higher risk of experiencing IPV compared to men, with nearly 1 in 7 men experiencing IPV [3,4]. It is particularly high for women who are married, where one in five married women experiencing some form of violence from their spouse [5]. The global prevalence of IPV averages around 30% [6]. In high-income countries, the prevalence of IPV is approximately 23% in [7], where in LMICs, such as Bangladesh, it can be as high as 75% [8,9]. These alarming prevalence rates of IPV raise concerns due to their potential for causing significant harm to women and communities.

Impacts of IPV on physical health include head injuries, hearing damage, bruises, broken bones, back and neck injuries, etc. [10], sometimes leading to fatal consequences. Specifically, the World Report on Violence and Health in 2012 reported that 40–70% of women victims were killed by their intimate partners [10]. Furthermore, IPV is found to be associated with severe mental health effects, such as behavioural problems, sleeping and eating disorders, depression, anxiety, Post Traumatic Stress Disorder (PTSD), self-harm, suicide attempts, and poor self-esteem [11,12]. The impacts extend beyond women's health, with long-term consequences observed in children of IPV victims, who are at risk of behavioural and emotional disturbances [10], and broader harm to communities, including loss of productivity and increased homelessness.

Globally, there has been an increased effort to raise awareness and understanding of violence against women, including IPV [10]. IPV has been found to be associated with several maladaptive individual and interpersonal factors such as gender inequality and norms on the acceptability of violence against women [13]. Other contributing factors of IPV include lower education levels, child marriage, family violence, childhood abuse, dissocial personality disorder, harmful alcohol use, toxic masculinity, and unemployment [13–16]. These factors tend to be more prevalent within LMICs, highlighting the urgent need of evidence-based interventions to reduce the prevalence of IPV and provide appropriate support for the victims [17].

Several systematic reviews have been conducted on IPV in LMICs. However, most studies have only focused on identifying risk factors for IPV [18] or evaluating interventions to prevent or address IPV [19]. To develop effective prevention and response strategies, it is imperative to understand both the contributing factors and the consequences of IPV. To the authors' knowledge, no effort has been made to aggregate and systematically review both factors associated with IPV and consequences of IPV against women in LMICs. This study aims to provide a more comprehensive understanding of this prevalent public health concern. By identifying associated risk factors and consequences of IPV against women in LMICs, this systematic review enhances understanding of IPV and assists in identifying areas for further investigation that can inform interventions and policies to assist in achieving the Sustainable Development Goal 5.2 by 2030.

## Materials and methods

### Registration

The RISM-P (Preferred Reporting Items for Systematic Reviews and Meta-Analyses Protocols) guidelines (Moher et al., 2015) were followed in designing and reporting this systematic review. Additionally, the review has been registered with PROSPERO (Ref. No. CRD42022342777).

### Literature search

A systematic review of the existing literature was conducted using three online databases: ProQuest Central, Web of Science, and Scopus. These databases are widely recognized for their coverage of a diverse range of disciplines and have been extensively used in public health

research. They also include a wide range of high-quality peer-reviewed journals that meet PRISMA standards. In addition to these databases, a manual search was conducted through Google Scholar to find further relevant studies. Since there is no library index language in the databases mentioned above, individualised keywords were used for the search strategy. The search terms for this systematic review included various combinations and variations of the following words and phrases: IPV, abuse, violence, women, marital, spousal, partnered, risk factors, consequences, impacts, interventions, and LMICs.

## Inclusion and exclusion criteria

Four main inclusion criteria were developed to select articles more closely aligned with the objective of this systematic review. A study was selected if it (i) focused on intimate partner violence; (ii) used data from women in LMICs (According to the World Bank [20], LMICs refer to those countries with Gross National Income (GNI) per capita between US$1,036 and US$4,045); (iii) discussed the factors associated with IPV and/or the consequences of IPV; (iv) used a quantitative or qualitative or mixed method research design and (v) peer-reviewed journal articles published in English between January 2010 and January 2022. Other types of publications like case studies, conference papers, dissertations, and policy reports were not included in this study.

## Selection process

In this review, four steps were followed in the selection process. In the first step, all peer-reviewed articles were initially screened by evaluating the title and abstract for potential inclusion by the first author. The titles and abstracts of the selected articles were then independently reviewed by the second authors. This rigorous screening process was essential to ensure that only the most appropriate articles were included in our systematic review. Following a discussion, the discrepancies were resolved, and the articles accepted by both reviewers based on the abstracts were then retained for the full-text review. Whenever necessary, the third reviewer resolved conflicts between the first two reviewers. Finally, the first author screened the full text of the selected articles, and all articles included in this review were confirmed by all authors.

## Data extraction

The initial database search was able to identify a total of 961 peer-reviewed articles, including 194 articles from ProQuest Central, 466 articles from Scopus and 301 articles from Web of Science. Additionally, 23 articles were found through a manual search using Google Scholar. These articles were uploaded to EndNote software, and the first author removed 102 duplicate articles. As a result, 167 articles were shortlisted for full-text review. After thorough consideration, the review team agreed on 30 unique articles to be included in this full-text review. Table 1 provides a detailed summary of each article. The PRISMA flow diagram presented in Fig 1 outlines the above search and review process.

## Quality assessment

The National Institutes of Health (NIH) Quality Assessment Tool [47] was used to assess the quality of the included studies. This tool evaluated the observational cohort and cross-sectional studies according to the following criteria: research question, study population, sufficient timeframe to see an effect, frequency of measure, different levels of the exposure of interest, sample size justification, outcome measures, blinding of outcome assessors, follow-up rate, statistical analyses, exposure measures and assessment, and study setting. In the assessment process,

**Table 1. Detailed summary of selected articles including quality rating.**

| | Authors | Purpose | Region | Design of Study | Data Source | Sample Size | Data Collection Method | Quality (NIH Quality Assessment Tool) |
|---|---|---|---|---|---|---|---|---|
| 1. | Barnett, Halligan [21] | To explore associations between IPV subtypes (emotional, physical and sexual) and child development (cognitive, language, and motor) | Sub-Saharan | Quantitative | Primary | 626 mothers | Personal Interview | Good |
| 2. | Bhowmik and Biswas [6] | To examine the relationship between attitudes of women toward accepting IPV and sociodemographic predictors | South-Asia | Quantitative | Secondary | 63 689 ever-married women | MICS data | Good |
| 3. | Bondade, Iyengar [22] | Determine IPV and psychiatric comorbidity in women with infertility | South-Asia | Quantitative | Primary | 100 infertile women | Personal Interview | Good |
| 4. | Coll, Ewerling [8] | To assess the prevalence and inequalities in recent psychological, physical, and sexual IPV among ever-partnered women | Multi LMICs | Quantitative | Secondary | 372 149 ever partnered women | 46 DHS data | Good |
| 5. | Hajian, Kasaeinia [23] | To predict the effect of resilience and stress coping styles on the likelihood of suicide attempts in females reporting spouse-related abuse | Middle East | Quantitative | Primary | 150 non-pregnant female victims of IPV | Personal Interview | Fair |
| 6. | Jina, Jewkes [24] | To estimate the prevalence of emotional abuse in intimate partnerships among young women in rural South Africa and to measure the association between lifetime experience of emotional abuse and adverse health outcomes | Sub-Saharan | Quantitative | Primary | 1293 ever-partnered women | Questionnaire | Fair |
| 7. | Jiwatram-Negrón, Lynn Murphy [25] | To examine the synergistic effect of substance use (injection drug use), intimate partner violence, and HIV (dubbed the "SAVA syndemic") on depression and suicidal thoughts among a sample of high-risk women in Kazakhstan | East-Asia and Pacific | Quantitative | Primary | 364 ever-partnered women | Personal Interview | Fair |
| 8. | Meekers, Pallin [14] | To examine the relationship between Bolivian women's experiences with physical, psychological, and sexual intimate partner violence and mental health outcomes | Latin America and Caribbean | Quantitative | Secondary | 10,119 married or cohabiting women | Bolivia 2008 DHS data | Good |
| 9. | Miller, Okoth [26] | To quantify the lifetime prevalence of IPV among women in two rural Kenyan communities, as well as factors associated with IPV | Sub-Saharan | Quantitative | Primary | 873 women | Questionnaire | Good |
| 10. | Mootz, Basaraba [27] | To quantify syndemic risk in women and test associations among exposure to armed conflict, HIV status, and syndemic risk (i.e., IPV, mental distress, and alcohol use). | Sub-Saharan | Quantitative | Primary | 605 women aged 13–49 years | Personal Interview | Fair |
| 11. | Pahn and Yang [28] | To investigate the association between the maternal experience of intimate partner violence (IPV) and children's behavioral problems | East-Asia and Pacific | Quantitative | Secondary | 980 Cambodian mothers who have children aged 6–12 years | Data of the National Survey on Women's Health and Life Experience | Poor |

(*Continued*)

**Table 1.** (Continued)

| | Authors | Purpose | Region | Design of Study | Data Source | Sample Size | Data Collection Method | Quality (NIH Quality Assessment Tool) |
|---|---|---|---|---|---|---|---|---|
| 12. | Sanni, Hudani [29] | To examine the prevalence and individual and societal factors associated with IPV among Egyptian women | Middle East | Quantitative | Secondary | 12,205 ever-married women between the ages of 15 to 49 years | Data from the 2005 and 2014 Egypt DHS | Good |
| 13. | Shaikh, Pearce [30] | To assess how a woman's enabling resources and risk factors influenced the association of her exposure to IPV | Middle East | Quantitative | Primary | 608 ever-married women | Personal Interview | Fair |
| 14. | Sharma, Vatsa [11] | To assess the association of IPV against women with their mental health status | South-Asia | Mixed | Primary | 827 ever-married women | Questionnaire | Good |
| 15. | Leight, Deyessa [31] | To identify the direct and indirect pathways that link IPV to postpartum depression in women belonging to different ethnic–national groups in Israel | Middle East | Quantitative | Primary | Jewish (n = 807) and Arab (n = 248) women | Questionnaire | Fair |
| 16. | Wagman, Donta [32] | To examine husbands' recent use of alcohol as a predictor of physical or sexual IPV against women within 6 months of childbirth | South-Asia | Quantitative | Primary | 1,038 postpartum women's | Personal Interview | Good |
| 17. | Amegbor and Rosenberg [33] | To examine the spatial variability of the relationship between women's post-secondary education and IPV | Sub-Saharan | Quantitative | Secondary | 18,506 women | 2016 Uganda DHS data | Good |
| 18. | John, Kapungu [34] | To examine the relationship between early child marriage and psychological well-being and assessed if IPV mediates this relationship among young women | Sub-Saharan | Quantitative | Secondary | 969 ever-married women | Data from a study estimating the economic costs of child marriage in Ethiopia conducted in 2016 | Fair |
| 19. | McClintock, Trego [35] | To examine the association between controlling behavior and IPV | Sub-Saharan | Quantitative | Secondary | 37,115 women | DHS data from eight countries in Sub-Sharan (between 2011 and 2015) | Good |
| 20. | Memiah, Ah Mu [36] | To determine the prevalence of IPV and other moderating factors associated with IPV | Sub-Saharan | Quantitative | Secondary | 3,028 women | 2014 Kenya DHS data | Poor |
| 21. | Oluwole, Onwumelu [37] | To determine the prevalence and predictors of lifetime IPV among women in an urban community in Lagos, Nigeria | Sub-Saharan | Quantitative | Primary | 400 women | Questionnaire | Good |
| 22. | Rogathi, Manongi [38] | To assess the relationship between IPV and postpartum depression among women attending antenatal services in Tanzania | Sub-Saharan | Quantitative | Primary | 1013 pregnant women | Personal Interview | Good |
| 23. | Soleimani, Ahmadi [39] | To analyse the association between women's mental health and physical, psychological and sexual IPV | Middle East | Quantitative | Secondary | 2091 married women | Data from a population-based survey conducted in 2015 in Rasht, Iran | Fair |
| 24. | Ibrahim, Ahmed [40] | To assess the incidence and risk factors of IPV during pregnancy among a sample of women from Egypt | Middle East | Quantitative | Primary | 1,857 women | Personal Interview | Fair |

*(Continued)*

**Table 1.** (Continued)

| | Authors | Purpose | Region | Design of Study | Data Source | Sample Size | Data Collection Method | Quality (NIH Quality Assessment Tool) |
|---|---|---|---|---|---|---|---|---|
| 25. | Ahinkorah, Dickson [41] | To examine the association between women's decision-making capacity and IPV among Women in Sub-Saharan Africa. | Sub-Saharan | Quantitative | Secondary | 84,486 | DHS data from 18 countries in Sub-Sharan (between 2010 and 2016) | Good |
| 26. | Koen, Wyatt [42] | To examine the association between antenatal IPV and subsequent low birth weight in a South African birth cohort | Sub-Saharan | Quantitative | Secondary | 263 mothers | Data from the Drakenstein Child Lung Health Study (DCLHS) | Good |
| 27. | Sabri, Renner [43] | To examine risk factors for severe physical intimate partner violence (IPV) and related injuries | South-Asia | Quantitative | Secondary | 64704 ever-married women | Data from 2005–2006 India National Family Health Survey | Good |
| 28. | Kouyoumdjian, Calzavara [44] | To quantify the association between IPV and incident HIV infection in women | Sub-Saharan | Quantitative | Secondary | 10252 sexually active women | Data from Rakai Community Cohort Study annual surveys between 2000 and 2009 | Good |
| 29. | Diamond-Smith, Conroy [45] | To explore the relationship between different levels of food insecurity (none, mild, moderate, severe) and three types of IPV: physical, sexual and emotional | South-Asia | Quantitative | Secondary | 3373 married women | Data from 2011 Nepal DHS | Good |
| 30. | Hayati, Högberg [46] | To examine associations between physical and sexual violence among rural Javanese Indonesian women and sociodemographic factors, husband's psychosocial and behavioral characteristics and attitudes toward violence and gender roles | East-Asia and Pacific | Quantitative | Primary | 765 women | Personal Interview | Good |

IPV = intimate partner violence; DHS = Demographic and Health Survey; MICS = Multiple Indicator Cluster Survey.

three independent authors rated the studies as "poor", "fair", or "good" based on the criteria described in the NIH Quality Assessment Tool. The articles that met less than six criteria were classified as "poor", those that met 6 to 10 criteria were classified as "fair", and those meeting more than 10 criteria were classified as "good". Consistency among the three reviewers' ratings was ensured, and any discrepancies were resolved by an independent fourth reviewer. This quality assessment was designed primarily to evaluate the methodological rigor and potential biases of the included studies. No studies were removed following quality assessment. The final assessments were confirmed by all the authors and the quality assessment results of each article are presented in S1 Table.

## Results

### Study characteristics

A detailed summary of the 30 included studies is presented in Table 1 which includes author's name and year of publication, purpose of the study, region in which study was undertaken, type of study design used, data source, sample size and category of participants, data collection method and quality of the study. This review refers to 30 studies conducted between 2010 to 2022 in several LMICs from five different regions, with only one study covering multiple

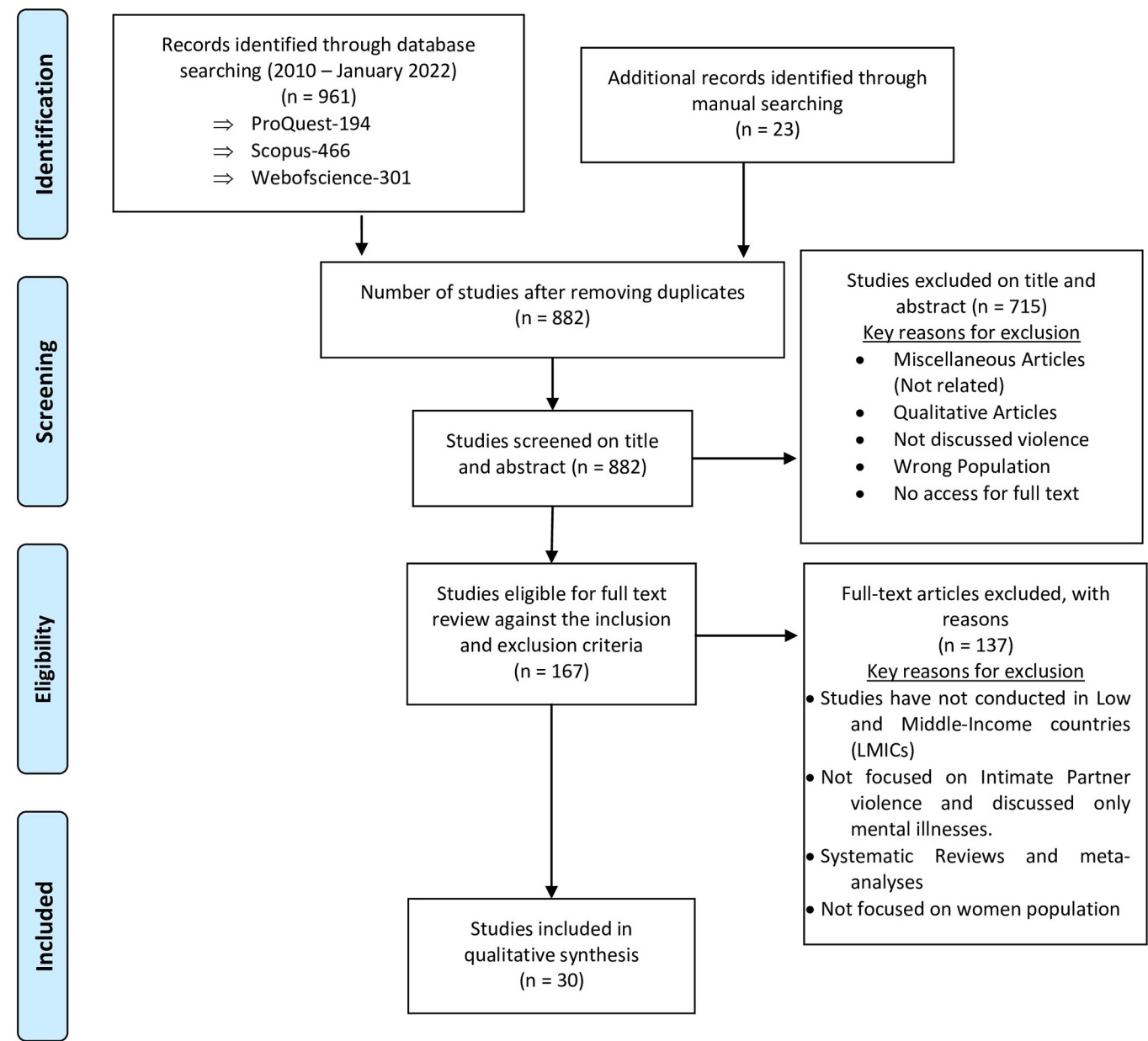

**Fig 1. PRISMA flow diagram of study selection.**

countries from more than one region [8]. The highest number of studies was carried out in Sub-Saharan Africa (13), followed by the Middle East (6), South Asia (6), East Asia and Pacific (3), Latin America and Caribbean (1). Most of the studies (n = 29) used a quantitative approach, while only one used a mixed-method design [11]. Half of the studies (n = 15) considered primary data while the other half used secondary data, including global national survey data such as DHS data, and Multiple Indicator Cluster Survey (MICS) data. The majority of studies based on primary data involved interviews with women (n = 10) and five studies collected primary data using survey questionnaires. In more than half of the studies (n = 16), data were analysed using Logistic regression models to identify the risk factors and the consequences of IPV. Because the included studies varied in design, sample size, risk factors,

consequences, objectives, and locations, this systematic review provided findings in the form of a qualitative analysis rather than a meta-analysis.

**Quality assessment.**   A total of twenty-one studies were rated as good, while nine studies were rated as fair, and two studies were rated as poor (Table 1). The works by Pahn and Yang [28] and Memiah, Ah Mu [36] received a "poor" rating due to several unmet criteria on the NIH Quality Assessment Tool. The major shortcomings of these two studies were their inability to specify and define the population correctly, and their failure to report participation rate, sample size justification, and assessor blinding status due to inherent cross-sectional design limitations. Despite these shortcomings, both studies had adequate power (n = 980 and n = 3028) and used cross-sectional self-reported opinion surveys. Hence, the authors decided to include them in the review. A detailed quality assessment of each study is presented in S1 Table.

## Risk factors associated with IPV

Risk factors of IPV identified through this review were categorised into four groups: demographic risk factors (such as age, residence, education level, religion, marital status, socio-economic status, and attitudes toward IPV), family risk factors (including childhood abuse, number of children, and extra-marital relationships), community-level factors (such as lack of social support) and behavioural risk factors (such as depression, alcohol/drug use, controlling behaviour, and help seeking behaviour).

**Demographic risk factors.**   This review found that seven demographic factors were significantly associated with IPV against women in LMICs. They are education status (n = 11), age (n = 9), economic status (n = 9), residence (n = 4), religion (n = 3), marital status (n = 2), and attitudes toward IPV (n = 3). The following section describes the influence of demographic risk factors of IPV identified in this review.

*Education status.* The results of eleven studies in this review show a significant association between women's educational level and IPV against women in LMICs. A total of six studies found that women with lower primary education levels or no education are at a higher risk of experiencing IPV [23,29,31,38,41,44]. By analysing Uganda's 2016 DHS data, Amegbor and Rosenberg [33] found a significant association between women's post-secondary education and a lower risk of IPV exposure.

Three studies have found a significant relationship between IPV against women in LMICs and their partner's education. Among these three studies, two studies demonstrated that women with partners who completed a primary or lower level of education were more likely to experience IPV than women with partners who completed secondary or higher levels of education [31,42]. Results obtained from the selected studies indicate that the years of formal education for the male partners were associated with experience of IPV for the female partners. Women whose husbands had less than 9 years of education were found to be at a higher risk of exposure to IPV [46] as compared to women whose husbands had more than 9 years of formal education. Research has also demonstrated that male partners without tertiary education are more likely to be perpetrators of IPV [37].

*Age.* Ten studies in the current review found a significant relationship between IPV and women's age in LMICs. Several studies indicated that women under 30 years of age are at a high risk of IPV [8,31,39,41,44]. Women who married at an early age (below 19 years) also had a higher risk of experiencing IPV than their counterparts who married later [36,40]. However, there are some exceptions with some studies indicating that younger women were less likely to experience IPV [23,42].

In two studies, the partner's age was also identified as a significant risk factor for IPV against women in LMICs. Memiah, Ah Mu [36], a study on women in Kenya, which

demonstrated that women with husbands or partners aged over 50 years had a slightly increased risk of experiencing IPV than those whose partners were younger than 29 years. Another study showed that Indonesian women were more likely to suffer sexual violence when their husbands were younger than 35 years [46].

*Economic status.* Nine studies in this review showed that IPV was significantly associated with economic status, wealthier women or women with a rich wealth index reported less IPV relative to the poorer women or women with poor and middle wealth indices in four studies [8,23,42,44].

Studies also showed that women's employment status is significantly associated with IPV against women in LMICs, with the risk of experiencing IPV higher for the women who were currently employed compared to those who were unemployed [23,42,44]. In contrast, two studies in this review determined that unemployed women had higher odds of experiencing IPV as compared to their counterparts [35,38]. When investigating inconsistencies related to employment status and IPV, it was found that educational disparities may contribute to the inconsistent results observed across different studies. Most of the women who were included in the studies that revealed that employed women were at increased risk of IPV had a primary education or no education. Conversely, most of the women who participated in studies showing that working women are less likely to develop IPV had tertiary or post-secondary education.

The financial stability of women is associated with their ability to make their own decisions [48]. It was demonstrated by two studies that IPV occurs more frequently among women who have decision-making capacity [42,46], while only one study considering 46 LMICs [8] found empowered women were less likely to experience IPV.

*Residence.* The relationship between residential area and risk of IPV was examined in three of the included studies, with findings indicating that living in rural area was associated with a higher risk of experiencing IPV among women [6,8]. Furthermore, among Indian women, it was observed that living in mega cities reduced the likelihood of IPV-related injuries by 29% compared to those living in rural areas [43].

*Religion.* Two studies on women in Sub-Saharan Africa region found that women belonging to other religion groups, having no religion, or being Christians were more likely to experience IPV compared to those who identified as Muslims [23,42]. Contrary to this, Sanni, Hudani [29] found that Egyptian women with an Islamic religious background were more likely to experienced IPV than the women who were Christians.

*Marital status.* Only two studies examined the relationship between marital status and IPV. In two studies, it has been shown that women who have been separated or divorced [43] or involved in a polygamous marriage are at a higher risk of IPV [26] as compared to those who are currently married, widowed, or in a monogamous marriage.

*Attitudes toward IPV justification.* Community attitudes and norms around the acceptability of violence against women were identified as risk factors for IPV in several studies. A study by Sabri, Renner [43] showed that Indian women who accept violence have a higher chance of suffering physical IPV and injuries caused by IPV. In a study of Indonesian women, Hayati, Högberg [46] found that women who agreed with the "good wife obey her husband" and "a man should show who is boss" were at a higher risk of experiencing sexual violence in relation to the women who disagreed. Further, this study determined that women who justified male violence were more likely to experience both physical and sexual violence. IPV was higher among Kenyan women when they believed their partners were justified in beating them than those who felt their partners were not justified in beating them [36].

**Family risk factors.** Six studies in this review demonstrated that various family risk factors were associated with IPV incidence among women in LMICs, such as childhood exposure

to violence, childbearing, and extra-marital relationships. Childhood exposure to violence was strongly positively associated with IPV against women in three studies [29,38,44]. The findings of a study of Nigerian women demonstrated that women who had witnessed parental violence in their childhood were more likely to experience IPV in their adult lives [37]. Moreover, among ever-married Indian women exposure to violence in childhood was significantly related to physical violence by their husbands [43]. Another study reported that Kenyan women exposed to violence in their girlhood were at a greater risk of experiencing IPV than those who were not [26].

Two studies also recognized having extra partner relationships as a risk factor for IPV against women in LMICs [23,38]. These two studies found that women who had multiple sexual partners other than their husbands or partners were more likely to be exposed to IPV. According to Memiah, Ah Mu [36], Kenyan women who received money, gifts or any favours in exchange for sexual activity were at higher risk of becoming IPV victims than women who did not accept such exchanges. Further, the study also reported that compared to the assertive Kenyan women, those who were not sexually assertive had a significant decrease in the odds of experiencing IPV.

Another risk for IPV identified in this review is the number of children given birth by a woman. In a study of Indian women, Sabri, Renner [43] found that women with larger number of children experienced more physical IPV and IPV related injuries.

**Community level risk factors.** Lack of social support was identified as a risk factor of IPV in a study conducted in Israel, indicating that women with limited social support had greater exposure to IPV [49].

**Behavioural risk factors.** In the present systematic review, six studies identified behavioural risk factors including husband's alcohol usage, controlling behaviours of both victims and perpetrators, and help seeking behaviour are associated with IPV against women in LMICs.

*Husband's Alcohol usage*: Sabri, Renner [43] found that Indian women's alcohol usage was not significantly associated with IPV. However, the study revealed that their husbands' alcohol usage was related to developing physical violence and severe IPV related injuries. In addition, four studies also found that a partner's adverse alcohol consumption resulted in a higher prevalence of IPV against women in LMICs [30,38,41,47].

*Controlling behaviour*. Controlling behaviour was found to be associated with IPV in several studies, indicating that women with negative controlling behaviours such as jealousy and unfaithfulness were more likely to experience any form of IPV [35]. In addition, two studies have shown that husbands' or partners' adverse characteristics including jealousy, suspicion, unfaithfulness, fighting with other men and emotionally and sexually abusive behaviours had also increased the likelihood of women's exposure to IPV [44,47].

*Help seeking behaviour*. Sabri, Renner [43] concluded that Indian women who sought formal or informal help were more likely to experience severe physical IPV than women who did not seek any help.

## Consequences of IPV

There were three types of consequences of IPV identified in various studies; mental health (such as depression, anxiety, Post Traumatic Stress Disorder (PTSD), Postpartum Depression (PPD), and suicidality), physical and sexual health (such as injuries, HIV and unplanned pregnancies) and impacts on children (such as children's nightmare and difficulty in child development).

**Mental health impacts.** Five different mental health impacts of IPV were identified in various studies; Depression (n = 8), Anxiety (n = 4), suicidality (n = 4), PPD (n = 2), PTSD

(n = 2), and hazardous drinking behaviour and drug use (n = 1). The following section describes the mental health consequences of IPV identified in this review.

*Depression.* Depression was identified as a significant consequence of IPV in eight studies. Among them, three studies found that women who experienced emotional IPV were more likely to report depression than women who experienced physical and/or sexual IPV [14,27,29]. One study also reported that depression was higher among drug-involved IPV suffering women than in drug-involved women who did not experience IPV in Kazakhstan [25]. Furthermore, depression was significantly higher among IPV victimised women who were exposed to armed conflict in Uganda [27]. A couple of studies demonstrated that married women who experienced IPV had higher odds of having depression symptoms [11,40].

*Anxiety.* Four studies in this review identified a positive association between IPV and anxiety among women in LMICs. The experience of IPV was associated with higher scores on the Hamilton Anxiety Rating Scale among infertile women Bangalore [22] and women who have experienced sexual or psychological IPV were more likely to experience anxiety symptoms [14,32,40].

*Suicidality.* Suicidality was reported as a consequence of IPV on women in LMICs in four of the reviewed studies. Sharma, Vatsa [11] found that married women exposed to IPV in Delhi were more likely to consider suicide than their counterparts who had not been exposed to IPV. In Kazakhstan, drug-injected HIV-positive women exposed to IPV had a 6-fold higher risk of reporting suicidal thoughts than women who did not report IPV or HIV [25]. Jina, Jewkes [24], who studied young African women, found that psychologically abused young women were more likely to commit suicide than those who experienced no violence. However, problem-oriented coping strategies and higher resilience were reported as protective factors for suicide attempts among IPV-exposed Iranian women [23].

*Postpartum Depression (PPD).* Two studies reported that IPV significantly increases the risk of PPD among women in LMICs. Compared to women who were not exposed to IPV during pregnancy, those who experienced at least one form of IPV during pregnancy were more than three times likely to have PPD [38]. Furthermore, women with no previous history of depression were at higher risk of experiencing PPD when exposed to psychological violence relative to women with a previous history of depression. Similarly, in an Israeli sample, IPV affected women with unplanned pregnancies were more likely to suffer PPD [49].

*Post-Traumatic Stress Disorder (PTSD).* Two of the studies identified PTSD as a consequence of IPV. A high prevalence of PTSD was found among IPV-affected infertile women in Bangalore [22], and most Iranian women exposed to IPV suffered from PTSD [23].

*Hazardous drinking behaviour and drug use.* Jina, Jewkes [24], reported that South African women who experienced physical and/or sexual IPV with emotional abuse were much more likely to engage in hazardous drinking behaviour and use illicit drugs.

**Physical and sexual health impacts.** This review identified the physical impacts of IPV against women in LMICs through three studies. A study by Ibrahim, Ahmed [40] concluded that in Egypt most of the pregnant women exposed to IPV reported that their pregnancy was unplanned. Further, women who experienced physical violence showed the highest risk of having adverse pregnancy outcomes including abortion, miscarriage, and preterm labor. In addition, this study identified that pregnant women who experienced IPV tended to have inflicted wounds including contused wounds, firearm injuries, and stab wounds. Sabri, Renner [43] identified that IPV suffering Indian women were highly associated with severe physical injuries. HIV was identified as a significant impact of IPV, and it is widely spread among women who experienced a longer duration of IPV and the women who exposed to more severe and frequent IPV [50].

**Impacts on children.** In this review, four studies examined the adverse effects of IPV on children in LMICs. One study found that South African women who had experienced maternal emotional IPV had lower cognitive, language, or motor composite scores for their children at age two [21]. Additionally, two-year-old children's motor scores were lower when their mothers were exposed to physical IPV. The relationship between IPV and child development was not mediated by maternal depression or alcohol consumption [21]. Using data from the National Survey on Cambodian women's health and life experience, Pahn and Yang [28] found that children with mothers who experienced IPV experienced nightmares, bedwetting, and timidity at a higher rate than those without that experience. Moreover, children whose mothers experienced physical violence had a higher rate of aggression. Two studies reported that IPV during pregnancy had adverse effects on newborn health, including lower birth weight, weight-for-age Z scores, fetal distress, and even fetal death [41,43].

Most of the studies included in this current review reported only the significant factors and consequences associated with IPV. However, associations between IPV and certain factors, including ownership of a mobile phone, the type of family (joint or nuclear), sexual assertiveness, number of children, and length of marriage were not found to be significant in some of the reviewed studies [6,25,29,36,41,47].

## Discussion

This systematic review has identified several risk factors associated with IPV against women in LMICs, such as a lower level of education, a younger age, a low wealth index, rural residential areas, childhood exposure to violence, extramarital relationships, the increased number of children, norms on acceptability of violence, a lack of social support, and adverse partner or husband characteristics. The review also identified depression, anxiety, PTSD, PPD, suicidality, severe physical injuries, unplanned pregnancies, HIV, and barriers to child development as common consequences of IPV.

The association between lower education attainment and IPV emerged as a consistent finding in the current review, with the majority of studies demonstrated this significant association. One possible explanation for this association is that lower education may limit women's economic, literacy and social resources [51,52], making them more vulnerable to IPV. Further, in many LMICs, factors such as prevailing gender norms within rural communities as well as limited access to quality education facilities can further contribute to a lower educational level among women, increasing their vulnerability to IPV [31,53,54]. Furthermore, it is believed that lower education levels may contribute to the acceptance of male dominance and control within relationships due to traditional gender norms. Further research should investigate the complex relationship between education, gender norms, and IPV in LMICs. Interventions that aim to increase educational opportunities for women and challenge harmful gender norms should be prioritized by policymakers and practitioners to prevent and respond to IPV.

The relationship between women's employment and IPV has been the focus of numerous studies, with some suggesting that women's employment status may be a potential risk factor for experiencing IPV [42,55]. While the evidence on the relationship between women's employment and IPV is mixed, it is clear that economic dependence and power imbalances within relationships play crucial roles. Some studies have found that women's employment is associated with a reduced risk of IPV [35,38]. Possibly, this is because employed women have greater bargaining power within their relationships. Thus, women can contribute in household decisions and have better control over household finances, as well as become more independent from their partners, which ultimately leads to greater autonomy [56,57]. With greater autonomy, women have the freedom to make their own choices rather be dependent on

their partners, which may allow them to leave abusive relationships and seek support [15]. However, women's employment may challenge traditional gender norms and power imbalances within their intimate relationships, increasing the likelihood of IPV [42,55,58]. Therefore, further research considering mixed-method study design are needed to gain a better understanding of the mechanisms by which employment may influence IPV.

In terms of IPV, higher socioeconomic status is a protective factor [23,42]. This appears to be due to the fact that women with greater economic resources may have greater access to services and support, which could reduce their likelihood of experiencing IPV. Further, the findings of this study indicated that older women were less likely to experience IPV than younger women, particularly in environments where child marriage is prevalent [16,59]. A possible explanation for this could be that families in low-income nations cannot afford to send their daughters to school, increasing the likelihood of early marriage [59]. Early marriages are often characterized by power imbalances in the relationship, with younger women being more susceptible to IPV [36,40]. Younger women are also found to be impacted by social norms, cultural expectations, gender discrimination, and financial vulnerabilities leading to fewer employment opportunities, lower social status, and limited decision-making power [60–63]. As a result, there is an imbalance of power within relationships, which increases the likelihood of young women being exposed to IPV. Therefore, older women may be more capable of negotiating and asserting their rights within the household, as well as being less likely to experience IPV. However, it is possible that older women report less IPV because of additional barriers, such as the stigma associated with IPV and societal expectations for long-term marriages [64]. Social norms that discourage openly discussing marital conflicts or seeking outside assistance may also discourage older women from reporting IPV incidents [64,65]. Consequently, IPV may be at a lower prevalence among older women as compared to younger women. Thus, interventions aimed at reducing IPV should be targeted at women in younger age groups, particularly those who are at risk of or have experienced child marriage. Additionally, older women also face different challenges, so interventions should also emphasize creating a safe environment for them to seek support and report IPV incidents.

Women's autonomy plays a major role in reducing IPV, as it empowers them to resist and escape abusive relationships by making decisions about their own lives, including their relationships, finances, and healthcare [66]. However, some researchers have suggested that IPV is more likely to occur among women who have decision-making capabilities [42,46]. The reason for this could be the fact that, in societies characterized by rigid gender norms and roles, women who challenge traditional gender roles and expectations may be considered violating cultural norms, which could increase their risk of IPV [63]. Alternatively, women with limited decision-making power are more vulnerable to IPV as they have less power to negotiate access to support services or leave abusive relationships [63]. To effectively prevent and address IPV in LMICs, it is crucial to explore and understand the complex and context-specific factors that influence the relationship between women's autonomy and IPV, such as cultural norms, economic dependence, and other relevant factors. Therefore, a well-designed qualitative or mixed-method study is necessary to delve deeper into these issues. Further, community-based programs including awareness campaigns need to be developed to promote gender equality and respect for women's rights.

The residential area is a significant socio-demographic factor linked to IPV in LMICs. The prevalence of IPV varies from rural to urban areas, with rural women experiencing a higher rate of IPV. One possible explanation to this is that women in rural areas are less likely to have access to appropriate services and information [6,67]. Previous research has indicated that exposure to IPV is more common among women in rural or remote areas than urban areas of a country. Several factors may contribute to this, including low socioeconomic status and

limited educational opportunities, which are common among the women reside in rural areas [53,68]. Additionally, in rural areas women lack access to social support and media, and they tend to accept social and religious norms that emphasise male dominance [6,8,31]. Media campaigns are identified as an effective awareness dissemination tool for women in LMICs [67]. Assisting women in remote areas with access to media and mobile networks can assist them in improving their awareness about IPV, resulting lowering the IPV prevalence.

Attitudes and norms related to unequal gender relationships are more common among women in LMICs. Based on this study, it was found that prevailing social norms and beliefs often justify IPV, which in turn results in its normalization and perpetuation. Acceptance of IPV may be influenced by factors such as gender inequality, low socioeconomic status, low educational attainment, traditional gender roles, and cultural norms that allow violence against women [9,69]. It is important to note that these attitudes can also have severe consequences for women, including physical and psychological harm, social stigma, as well as limited access to education, resources, and services. Community-based educational programs, empowerment initiatives for women, and engagement with key stakeholders including religious and cultural leaders are necessary in minimizing gender inequality [70].

In line with previous research, the current review demonstrated that witnessing and experiencing abuse as a child and childhood trauma conditions may result in a higher chance of having violent relationships in adulthood [29,71,72]. Childhood exposure to violence can result in a wide range of negative outcomes, including trauma, mental health issues, impaired social, emotional development, and interpersonal difficulties [73,74]. Consequently, individuals who have been exposed to violent behaviour as a child may find it difficult to establish trust, intimacy, or resolve conflict in their adult relationships, which may increase the likelihood of experiencing or perpetrating IPV [75,76]. In addition, the intergenerational transmission of violence theory shows that individuals who have been exposed to violence during their early childhood are more likely to continue their violent behaviour in response to conflict and stress in their later lives [77,78]. Thus, it is possible for them to repeat these patterns of aggression and control in their intimate relationships, becoming both victims and perpetrators of IPV [76,79]. A comprehensive understanding of how childhood trauma affects the development of violent relationships is therefore necessary for effective prevention and intervention strategies.

Furthermore, IPV is more likely to occur when women have more than one sexual partner in addition to their husband. The main reason may be that infidelity can cause relationships to deteriorate, leading to conflict, which in turn leads to divorce and/or separation with severe IPV [23,38,44]. The husband's undesirable traits and troublesome conduct, such as excessive drinking, as well as negative control behaviours (jealousy, suspicion, unfaithfulness, etc.), can also contribute to impulsive reactions to verbal arguments or conflicts between husband and wife, leading to IPV [30,38,71,80]. The findings indicate that social and community-based interventions are necessary by targeting the vulnerable group of women who witnessed violence during childhood, experienced extra marital relationships, or whose husbands exhibited troublesome behaviours.

IPV has widespread and long-lasting consequences for women and families [8], leading to serious short-term and long-term negative psychological and physical impacts [81] on women as well as community. These effects include damage to a person's health, long-term harm to children by suffering a range of behavioural and emotional disturbances and harm to communities by loss of productivity and increased homelessness [10]. The current review supports and provides evidence of these consequences, emphasizing that IPV is not only a burden for women but also for their families and the community, affecting their overall well-being and productivity. This systematic review highlights the need for interventions that mitigate IPV

impacts. These interventions may include providing safe spaces for victims to seek support and counselling, as well as improving access to mental health support services [75–77]. Furthermore, evidence-based interventions, such as trauma-informed care and cognitive-behavioural therapy interventions, have shown effectiveness in reducing IPV impacts and promoting survivor well-being [77–81].

According to stress theory, IPV can have significant mental health impacts on women, including depression, post-traumatic stress disorder (PTSD), and anxiety [82,83]. The current review provides evidence that these IPV outcomes can lead to fatigue, disruption of daily activities, and getting scared easily [84–86]. PTSD is a common mental health problem reported by women who have experienced IPV, and it can make victims feel helpless and fearful for their safety [87–89]. The stress theory also suggests that the ongoing stress and trauma due to IPV can lead to long-term changes in the brain and nervous systems [90]. To cope with mental health disorders caused by IPV, some women may develop harmful maladaptive coping strategies, such as suicide [11,27,91]. Furthermore, a variety of maladaptive coping strategies may result, such as self-harm, substance abuse, hazardous drinking behaviour, and eating disorders [11,12,33,86,87]. Additionally, women with higher education levels are better equipped to protect themselves against the consequences of poor mental health, as they possess knowledge of available resources, propensity to make decisions, and the ability to cope [11,92]. To address the psychological consequences of IPV, a comprehensive program of mental health services and counselling should be provided to women who have experienced IPV. It is necessary to develop innovative interventions that emphasize education and community awareness programs in order to empower women and communities to combat IPV as well as to provide individuals with the necessary knowledge and tools to protect their mental health. Furthermore, support from family, friends, and community organizations can provide significant support to women in coping with the psychological effects of IPV [17].

A review of studies in the current study demonstrated the negative effects of IPV on women's physical and sexual health as well as their children's health. It has been reported that physical IPV causes minor injuries to severe health conditions, including severe injuries, and chronic pain [93]. There is a significant impact on lives of IPV victims who experience sexual and reproductive health problems, including HIV infection [88,94,95]. These individuals often experience considerable suffering as well as many physical, emotional, and social difficulties. As well as being destructive and difficult to deal with for adults, IPV also poses challenges to children, resulting in developmental problems [24,43], psychological problems (nightmares, bedwetting, timidity, fetal distress), behavioural problems [28] and even fetal deaths after being exposed to IPV by their mothers [41,43]. The impacts of IPV on children can last into adulthood, perpetuating the cycle of violence. In order to address these physical and sexual health consequences of IPV, interventions should focus on providing emergency medical services as well as developing community-based programs that promote healthy relationships. Further, providing legal protection to women who have experienced IPV may help prevent future incidents of abuse and provide a sense of safety and security [96].

In summary, this study findings revealed that IPV affects the physical and mental health of women as well as society in a significant way, and it adversely affects women's productivity and ability to care for themselves and their families. IPV victims in LMICs need urgent assistance with more access to social support in order to improve their mental health and quality of life.

## Strengths and limitations

To the authors' knowledge, the current review is the first attempt to aggregate both causes and consequences of IPV against women in LMICs. While providing useful information about IPV

against women in LMIC, the current review does have some limitations. First, this review was limited to peer-reviewed journal articles published in the English language, and studies published in other languages or in non-peer-reviewed sources were not included. Second, the focus of our systematic review is primarily on heterosexual women who are victims of IPV. Therefore, our findings may not be generalizable to individuals with diverse sexual preferences and gender identities. Third, due to the heterogeneity of the studies, it was difficult to conduct a meta-analysis to incorporate statistical methods which would provide a more precise estimate of the influence of the risk factors of IPV along with their effect size, and only qualitative analysis was conducted. Forth, the current review provides only a preliminary overview and indication of the potential risk factors and consequences of IPV but does not have sufficient data to examine the interactions between various factors (demographic, family, behavioural, and community) in their contribution to IPV. Fifth, this review only focuses on factors and consequences associated with IPV and does not include studies based on interventions. Therefore, this may not provide a comprehensive picture of the effectiveness of interventions in addressing and preventing IPV. Consequently, it is necessary to conduct properly designed research based on appropriate theoretical models to better understand the interfaces of risk factors. Another limitation of this review is the relatively small number of articles available for inclusion especially for some of the factors or consequences, which may affect the generalizability of our results. Finally, as the reviewed studies used cross-sectional data, no causal inferences can be made. These limitations should be taken into consideration while interpreting the findings of the current review.

## Conclusions

The high prevalence of IPV was found to be significantly associated with a variety of demographic, family, community, and behavioural risk factors which emphasizes the complex nature of this public health issue. IPV has adverse effects on women, children, families, and society as a whole. Therefore, to effectively address IPV and it's impact, it is necessary to develop appropriate interventions based on these risk factors and consequences. The implementation of policy driven and appropriate interventions should be guided by a thorough understanding of IPV dynamics and tailored to the unique needs of different populations. Accordingly, this study highlights the need for further research to achieve more national level up to date data and critical analysis to explore the underlying factors and consequences of IPV, which in turns may facilitate more productive, preventive and intervention efforts in LMICs, particularly in South Asia, where IPV is particularly prevalent. Effective intervention programs and interference from social advocacy groups by targeting vulnerable group of women can also be considered to achieve SDG 5.2 by 2030.

### Implications for intervention and policy

As a result of this review, some implications for intervention and policy have been recommended. First, prevention policy and intervention programs can be developed to minimize the proven risk factors identified in this review (such as demographic, family, community, and behavioural risk factors) to prevent IPV. Secondly, a community-based intervention can be developed to increase women's awareness, education, mental health, and safety. Third, IPV victimized women in LMICs can be provided more psychological support. As a fourth point, most LMICs have male-dominated societies with limited power for women, so policies need to be developed to increase women's empowerment at all levels, which could indirectly help to reduce IPV. Finally, social support services and prevention and treatment programs need to be started at an early stage for the victims to address IPV issues by targeting the most vulnerable

group of women (such as young, uneducated women with low socioeconomic background) in LMICs.

## Recommendation for further research

The IPV prevalence rate was found to be relatively high in South Asian LMICs. However, this review highlights a significant gap in the existing literature, as only a few studies have been undertaken in this region despite the high prevalence of IPV. From this review, it is clear that, to date, only a few studies have been conducted to explore women's attitudes toward IPV in LMICs. Thus, there is a need to conduct rigorous large-scale population-based studies that explore women's attitude toward IPV in LMICs along with the risk factors and consequences of such violence on women's mental and physical health, the marital relationship, children, families, communities, and wider society. Based on this study findings, it can be concluded that education plays a significant role in reducing IPV. Nevertheless, it is important to note that education plays a mediation role as it often indirectly related to other contributing factors of IPV, such as gender inequality, women's empowerment, and cultural norms. Therefore, further research is needed to explore the complex interrelation between education and these factors and how they collectively influence the prevalence and consequences of IPV in LMICs. This includes exploring the path analysis by which level of education influence IPV, and how they interact with other socio-demographic determinants. Additionally, future research should focus on developing and evaluating educational interventions and empowerment programs designed to reach vulnerable individuals, families, and communities. The aim of these

**Table 2. Critical findings and implications.**

| Critical Findings | |
| --- | --- |
| **Outcome** | **Summary** |
| Risk Factors of IPV | • In LMICs, lower levels of education, marriage at a young age, poor wealth indices, and rural residential areas were identified as demographic factors which contributed to IPV risk among women. <br>• Witnessing and experiencing abuse as a child, having more children in the house, and having a husband with adverse characteristics such as hazardous drinking, and controlling behaviour (jealousy, suspicion, unfaithfulness, etc.) were significant family and behavioural risk factors for IPV among women in LMICs. <br>• Among community-level risk factors for IPV, a lack of social support and acceptance of norms promoting male dominance were found. |
| Impacts of IPV | • Depression, PTSD, anxiety, and suicidality were common mental health problems reported by women who have experienced IPV in LMICs.• Among LMICs, severe physical injuries, unplanned pregnancies, and sexually transmitted infections such as HIV were associated with IPV.• Having child developmental and psychological problems (such as lower cognitive score, lower birth weight, nightmares, and fetal distress) were identified as societal impacts of IPV. |
| **Implications** | |
| Implications for practice | • Community-based intervention can be developed to increase women's awareness, education, mental health and safety. <br>• Provide more psychological support <br>• Social support services and prevention and treatment programs need to be started at an early stage |
| Implications for policy | • Prevention policy can be developed to minimize the proven risk factors identified in this review |
| Implications for research | • Explore women's attitude toward IPV in LMICs along with the risk factors and consequences of such violence on women's mental and physical health, the marital relationship, children, families, communities and wider society <br>• Qualitative and mixed method studies linking appropriate theoretical model to understand the perceptions, subjective feelings, and experiences of IPV among women in LMICs, especially among Asian women |

interventions should be to promote empowered women through education and gender equality and enhance mental health and well-being. There is also a need to conduct a large scale qualitative and mixed method studies linking appropriate theoretical model to understand the perceptions, subjective feelings, and experiences of IPV among women in LMICs, especially among Asian women. A comprehensive summary of the main results is presented in Table 2 for ease of understanding.

## Supporting information

**S1 Checklist. PRISMA checklist.**
(DOC)

**S1 Table. Detailed quality assessment of the selected studies.**
(DOCX)

## Acknowledgments

The review was conducted as a part of the PhD study at the Swinburne University of Technology in Australia. The authors would like to thank M R S Sanjeewa for assisting in the initial screening of the abstracts of the articles.

## Author Contributions

**Conceptualization:** Lakma Gunarathne, Jahar Bhowmik, Pragalathan Apputhurai, Maja Nedeljkovic.

**Data curation:** Lakma Gunarathne.

**Formal analysis:** Lakma Gunarathne, Jahar Bhowmik, Pragalathan Apputhurai.

**Investigation:** Lakma Gunarathne, Jahar Bhowmik, Maja Nedeljkovic.

**Methodology:** Lakma Gunarathne, Jahar Bhowmik, Pragalathan Apputhurai.

**Project administration:** Lakma Gunarathne, Jahar Bhowmik.

**Resources:** Jahar Bhowmik.

**Supervision:** Jahar Bhowmik.

**Validation:** Lakma Gunarathne, Jahar Bhowmik, Pragalathan Apputhurai, Maja Nedeljkovic.

**Visualization:** Lakma Gunarathne.

**Writing – original draft:** Lakma Gunarathne.

**Writing – review & editing:** Lakma Gunarathne, Jahar Bhowmik, Pragalathan Apputhurai, Maja Nedeljkovic.

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
