## [Decision Letter · Decision Letter 0]

22 May 2023

PONE-D-23-08947Factors and Consequences associated with Intimate Partner Violence against Women in Low-and Middle-Income Countries: A Systematic ReviewPLOS ONE

Dear Dr. Gunarathne,

Thank you for submitting your manuscript to PLOS ONE. After careful consideration, we feel that it has merit but does not fully meet PLOS ONE’s publication criteria as it currently stands. Therefore, we invite you to submit a revised version of the manuscript that addresses the points raised during the review process.

We look forward to receiving your revised manuscript.

Kind regards,

Abraham Salinas-Miranda, MD, PhD

Academic Editor

PLOS ONE

Journal Requirements:

Additional Editor Comments (if provided):

Dear authors: two independent reviewers have concur that revisions are needed before the article can be accepted. This Academic Editor agrees with their comments and recommends major revisions including fixing grammatical issues, considering expansion to an additional bibliographic database, and the other comments

Reviewers' comments:

Reviewer's Responses to Questions

**Comments to the Author**

1. Is the manuscript technically sound, and do the data support the conclusions?

Reviewer #1: Yes

Reviewer #2: Partly

2. Has the statistical analysis been performed appropriately and rigorously? 

Reviewer #1: N/A

Reviewer #2: N/A

3. Have the authors made all data underlying the findings in their manuscript fully available?

Reviewer #1: Yes

Reviewer #2: Yes

4. Is the manuscript presented in an intelligible fashion and written in standard English?

Reviewer #1: Yes

Reviewer #2: No

5. Review Comments to the Author

Reviewer #1: Thank you for giving me the opportunity to review this paper. It presents an important systematic review of literature related to IPV and draws some important conclusions relevant for evidence-based IPV interventions.

Introduction

-Overall this introduction could benefit from revision and editing to make it more concise. There is a fair bit of repetition with the phrasing and statistics around the prevalence and impact of IPV and this section would benefit from being more succinct.

-On Line 103 you note that there are not current systematic reviews of “factors and consequences of IPV against married women in LMICs” and in Line 105-106 you note “By finding associated risk factors and consequences of IPV in LMICs, this systematic review enhances understanding of IPV…” I think it is very important to be crystal clear if you were only looking at studies of married women or all women who experienced IPV?

-Can you address in the introduction why the systematic review is just focused on women and not men who might also experience IPV? This may come out in limitations but as the introduction provides an overview of IPV and it’s consequences it would be helpful to address the fact that male victims of IPV are often not included in our discussions of IPV.

Quality Assessment

-It is not clear why the NIH Quality Assessment Tool was used to evaluate the included studies. Please explain why this tool was selected.

-Please clarify that all studies were ranked using the quality assessment tool but that rankings on the quality assessment tool did not affect whether the study was included in the systematic review. I think that is what the process was but it is not clear in the current section.

Results

oLine 224-228 is sharing a very important finding but it is very hard to follow the sentence structure. Please revise and perhaps divide into two sentences to make more clear.

oLine 277-279. Please rephrase “at a greater risk of developing IPV” as IPV is not something that is developed but rather experienced.

Discussion

-Line 431-440: The analysis of how economic status and age may impact the prevalence of IPV in LMICs being linked to girl children not going to school and being married young seems a bit reductive. Are there other factors that emerged from the literature review related to risk factors and the age of victims that could also be explored such as decision making power, employment, social standing etc.?

-Line 478-481: The link between the exposure to childhood trauma and negative outcomes as an adult is well understood and there is extensive literature on this. I would like to see this section more robustly developed to cite to this literature and discuss why the findings on this from the systematic review is important

-Line 481: I would suggest making the section on more than one partner it’s own paragraph

-Line 498-500: The recommendations to mitigate the impacts of IPV from this systematic review are important. I would love to see these recommendations more built out with additional citation from other literature and more detail about what types of interventions are indicated based on the findings of this specific literature review. Building this out will be very important for practitioners who want to cite this article when proposing IPV interventions.

-Line 507-509: I would delete “potentially” from the sentence as suicide is harmful in all instances. Additionally, I would welcome inclusion of additional maladaptive coping strategies observed other than suicide.

-Line 517-518: I would suggest rephrasing this sentence as it implies that those who are HIV positive have “unbearable lives” which can be stereotyping for those in the HIV positive community and should be avoided.

Strengths and Limitations

-I welcome seeing the analysis of how the search terms focused on martial/spousal relationships may have limited the study to understanding IPV against women in that population. It would also be helpful to see here in the limitations some discussion of how the small number of articles may impact the conclusions drawn. For some of the factors or consequences that were identified they were only listed in a few of the studies included which limits the readers ability to understand these factors as inclusive.

-Also, can you include a limitation on the fact that the study only focused on heterosexual women victims of IPV? And does not seem to look at additional vulnerabilities with intersectional identities which seems important to consider in future research

Conclusions

-The conclusion section could benefit from revision to strengthen the final conclusions drawn and the call to action. The recommendation for a large-scale population based study in South Asia does not to me seem linked to the findings and discussion which seemed to focus more on the need for more nuanced understandings of the factors related to IPV and how to construct interventions as a whole. The study does not seem to indicate a lack of data—rather a lack of detailed analysis of factors. I would expect to see the conclusions and recommendations for this paper more focused in this area and therefore suggest a rethinking and reframing of the conclusion.

Reviewer #2: Thank you for the opportunity to review the revised manuscript titled, “Factors and consequences associated with intimate partner violence against women in low- and middle-income countries: A systematic review” (PONE-D-23-08947). This study addresses an important gap in the literature – systematic reviews of IPV in LMICs are sorely needed. There are unfortunately several significant flaws in the author’s approach and description of methodology and results that will require substantial revision.

1.The appropriate reference for the WHO definition of IPV should be the World Report on Violence and Health (Krug, Dahlberg, Mercy, Zwi, & Lozano, 2002). The authors instead cite a WHO webpage, which itself reference the World Report (pages 2-3, reference 1) https://apps.who.int/iris/bitstream/handle/10665/42495/9241545615_eng.pdf

2.There are a number of grammatical issues throughout the manuscript that make some sentences difficult to follow (especially due to several sentence fragments and some run-on sentences). I recommend working with a strong language editor to recommend revisions to the paper.

3.The methods section indicates that only 3 databases were searched. Though this meets the most basic level of standards for high-quality systematic review, it barely does so. I have some concerns that several large databases were not included and the search may have therefore missed critical literature. For example, why were standard resources such as MEDLINE/PubMed, CABI’s Global Health, Social Science Research Network (SSRN), and CABI’s Global Health not included?

a.On page 6, line 146 the authors refer to a “manual search using Google Scholar,” but this is not described in the literature search methodology above. The fact that the Google Search did result in additional studies included suggests that only restricting the search to 3 databases could have resulted in significant missed literature.

4.More detail is needed regarding how the inclusion criteria were operationalized. For example, were studies included only if they presented data from adult women ages 18 and older? What about studies that also included adolescent girls? What about partnered but unmarried women?

5.The methods section does not describe any protocol or procedures to search the grey literature, such as searching websites for reports from multilateral and bilateral organizations and other grey literature. This omission very likely introduced bias in the systematic review approach taken by the authors.

6.In the results section, the authors summarize significant associations noted in the included studies. However, no mention is made of null results. Did all of the studies find only significant associations, or did the authors only selectively report significant associations? It seems important to note null results as they indicate lack of consistent patterns of findings in the literature.

7.For topics where the authors report inconsistent findings in the literature, it would be helpful if the authors included at least some detail to be able to assess what factors might account for inconsistencies. For example, the authors refer to some studies that found higher risk of IPV among women who were employed, but two studies found unemployed women had higher risk of IPV (lines 246-249). Were the studies from different geographic regions? Were the women’s ages different? Is there any information from the studies that can shed light on these patterns?

8.Similarly, I urge the authors to ensure the comparison or reference group is clear in their summary of findings. For example, the statements about evidence related to marital status (lines 266-268): is being separated or divorced and in a polygamous marriage associated with higher risk of IPV compared to currently-married women? Never-married women? Some other comparison?

9.I disagree with the author’s characterization of literature that assesses women’s individual attitudes about IPV as reflecting “community level risk factors” (lines 291-301). Studies that measure a woman’s attitudes or beliefs about violence reflect individual risk factors, not community risk factors. Those individual attitudes may be influenced by community norms and values. However, if the studies did not measure community norms and values and instead measured individual attitudes, that reflect individual risk factors nonetheless.

10.I also object to the author’s characterization of social support as a “protective factor” by virtue of the fact that its absence was found to be associated with higher risk of IPV in one study (lines 302-303). The two are not necessarily interchangeable. A more accurate assessment would be to indicate that lack of social support is a risk factor for IPV.

11.Similarly, the description of the findings from the Sabri, Renner study (lines 304-305) bear some consideration. Did the study find that women who had more severe physical IPV were more likely to seek formal or informal help? If so, that does not suggest that help-seeking is a risk factor for IPV. Some elaboration of the findings of this particular study are warranted.

12.The summary of the literature for “behavioural risk factors” appears to confuse risk factors with consequences. Several of the studies summarized appear to refer to what might have been mental health impacts of IPV rather than risk factors. This should be clarified.

13.I am also inclined to object to the use of the term “behavioural risk factors” to refer to phenomena like depression and other mental health issues. These are not always characterized behaviourally.

14.The authors assess study quality of the included studies and provide that information in a brief table, but study quality information does not appear to factor into the summary or conclusions. How was study quality taken into account in synthesizing research findings? Did it inform the authors in assessing consistencies or inconsistencies in the literature?

15.More detail could be provided regarding the gaps in the research – what are the areas that need further investigation? There is a very brief paragraph at the end that provides a somewhat perfunctory set of recommendations for future research. More could be said that is directly informed by the findings of the systematic review.

16.I was disappointed to find that the manuscript does not include a summary table with the overall findings related to risk factors and consequences. Table 1 provides an overview of the studies but it does not include summary findings, which seems to be the most important information.

6. PLOS authors have the option to publish the peer review history of their article (what does this mean?). If published, this will include your full peer review and any attached files.

Reviewer #1: No

Reviewer #2: No

---

## [Author Response · Author response to Decision Letter 0]

22 Jun 2023

We have addressed reviewers’ feedback one by one, and it can be found in the file name 'Response to Reviewers'.

---

## [Decision Letter · Decision Letter 1]

31 Aug 2023

PONE-D-23-08947R1Factors and Consequences associated with Intimate Partner Violence against Women in Low-and Middle-Income Countries: A Systematic ReviewPLOS ONE

Dear Dr. Gunarathne,

Thank you for submitting your manuscript to PLOS ONE. After careful consideration, we feel that it has merit but does not fully meet PLOS ONE’s publication criteria as it currently stands. Therefore, we invite you to submit a revised version of the manuscript that addresses the points raised during the review process.

ACADEMIC EDITOR: Dear authors: Thank you for addressing all the previous revisions requested. The reviewers identified some revisions that still need to be made including clarifications (line by line listed below) in several sentences and the issue of behavioral risk factors equated as outcomes. Please add an explanation when is missing in the segments requested by reviewer 1 and 2 or your argumentation if you do not agree with reviewers. 

We look forward to receiving your revised manuscript.

Kind regards,

Abraham Salinas-Miranda, MD, PhD

Academic Editor

PLOS ONE

Journal Requirements:

Additional Editor Comments (if provided):

The manuscript was vastly improved, but there are still several issues that were identified by the reviewers in all sections of the paper (introduction, methods, results, and conclusions). Please address each of these and resubmit as soon as possible.

Reviewers' comments:

Reviewer's Responses to Questions

**Comments to the Author**

1. If the authors have adequately addressed your comments raised in a previous round of review and you feel that this manuscript is now acceptable for publication, you may indicate that here to bypass the “Comments to the Author” section, enter your conflict of interest statement in the “Confidential to Editor” section, and submit your "Accept" recommendation.

Reviewer #1: (No Response)

Reviewer #3: (No Response)

2. Is the manuscript technically sound, and do the data support the conclusions?

Reviewer #1: Yes

Reviewer #3: Partly

3. Has the statistical analysis been performed appropriately and rigorously? 

Reviewer #1: Yes

Reviewer #3: N/A

4. Have the authors made all data underlying the findings in their manuscript fully available?

Reviewer #1: Yes

Reviewer #3: Yes

5. Is the manuscript presented in an intelligible fashion and written in standard English?

Reviewer #1: Yes

Reviewer #3: Yes

6. Review Comments to the Author

Reviewer #1: Thank you for the work to revise the manuscript and address the comments provided. The manuscript is improved but would benefit from further revision to strengthen the paper.

Abstract

• Line 28: Please clarify the statement “relatively high prevalence rates” as it is not clear what you are comparing this to? Relative to what? I would suggest choosing more precise language

• Line 29: should it read “within” LMIC’s rather than “among”?

• Line 39: You state “The findings indicate alarmingly high prevalence of IPV against women in many LMICs” This is confusing because it is not clear what this is being compared to? i.e. it is alarmingly high compared to what? Additionally quantifying the prevalence of IPV is not an explicit aim of this systematic review and therefore this sentence does not make sense to include in the abstract and is misleading to the study findings. Suggest rephrasing.

Introduction

• Overall the Introduction is still quite repetitive and not well organized. The topics covered in the introduction jump around significantly and it would overall benefit from additional editing

• Line 50: “IPV is known as” should read “defined as” as you are sharing a definition

• Line 61-62: “Women are at a higher risk of experiencing IPV compared to men” To make this statement the rate of IPV of risk for men needs to be cited to, as of now it’s not clear what you’re comparing to

• Line 93-96: This sentence is very confusing. Rephrase

Materials and Methods

• Line 109: It would be helpful to include more information about why these data bases were selected and why only 3 were used.

• Line 158: Please state explicitly that no studies were removed following the quality assessment

Results

• Line 242: should read “experiencing”

• Line 382: Is the reference to women from the country of South Africa or the southern part of the African continent? Not sure if south not being capitalized is a typo or not?

Discussion:

• Overall, the Discussion could benefit from revision with a particular focus on how risk factors may interact throughout the analysis. For example, I note on Line 408-410 that in the list of reasons why lower education may lead to IPV the other risk factors related to environment and access to education in the first place are not integrated into the discussion and analysis. I believe that there are many interactions between both the risk factors and consequences identified in the analysis and the paper would be greatly strengthened by including more nuanced discussion of these interactions.

• Line 438-447: In this paragraph it may be helpful to also consider why older women, if they experience IPV may not say anything or report it? What role might stigma or social norms play in less reporting of IPV by older women?

• Line 463-473: In discussion of the area where women live I don’t see discussion of economics or educational level, as they may be tied to living in a more rural location discussed though I would imagine that that could also have an impact on the risk of IPV in rural areas?

• Line 499: Was the research focused on women who have partners outside of marriage or men with multiple partners? The phrasing of this sentence is confusing to me

• Line 502 and 507: “husbands with adverse characteristics” is a weird phrasing to me. I would suggest rephrasing

Reviewer #3: The authors addressed most of the revisions adequately, including revising the introduction, explaining the methods (e.g., risk of bias tool), results, and limitations. However, this reviewer does not agree with the authors' response to reviewer #2 regarding the following:

COMMENT 1: From the previous revision: "Reviewer’s comment not addressed: 12. The summary of the literature for “behavioural risk factors” appears to confuse risk factors with consequences. Several of the studies summarized appear to refer to what might have been mental health impacts of IPV rather than risk factors. This should be clarified."

Considering the previous review comment, the Conclusion section on "Behavioural risk factors" (lines 356-378) needs to be revised. The only behaviors that appear to have some evidence for risk factors for experiencing IPV appear to be "partner's controlling behaviors" and "women help-seeking behaviors". Behaviors such as "women's alcohol use" and "psychological well-being" (not a behavior, but a mental health state) are not supported by study findings as risk factors.

The authors wrote a detailed explanation for why risk factors such as psychological well-being factors were not outcomes but risks. Nevertheless, the articles cited examined the directionality of IPV as risk to mental health outcomes (e.g., depression). For instance, in "Behavioural risk factors > Psychological well-being" (Lines 367-369), the paper cites reference 27 and 36. The data from these studies do not support psychological well-being as risk factor but as outcome as follows:

Ref. 27: Jiwatram-Negrón, T., Michalopoulos, L. M., & El-Bassel, N. (2018). The syndemic effect of injection drug use, intimate partner violence, and HIV on mental health among drug-involved women in Kazakhstan. Global social welfare : research, policy & practice, 5(2), 71–81. https://doi.org/10.1007/s40609-018-0112-1. This reference (reference number 27) found that the syndemic of IPV, substance abuse, and HIV increased 15.5 fold odds of reporting depression. Not the other way around. IPV was not significantly correlated with injected drugs (Jiwatram-Negron's Table 2: r=-0.047 between IPV and injected drugs). On Table 3 of the same article, victim's binge drinking was not significantly associated with depression and suicide either.

In the subsection: "Behavioural risk factors > Alcohol usage" (Lines 360-361), the paper cites reference 27 as follows: "Alcohol usage: Hazardous drinking among women was found to be significantly associated with physical and/or sexual IPV experienced by south African women [27]." This sentence must be deleted as it is not factual.

The data from ref 27 do not support women's substance abuse as risk factor for IPV. Jiwatram-Negron only reported correlation, which was not significant. Correlation should not be confused with risk factor (i.e., the word risk factor implies a directionality). When Jiwatram and colleagues discussed women's substance abuse and IPV, they used "injected drugs" not "hazardous drinking" like it was mentioned in the paper here. Jiwatram-Negron and colleagues used the lens of syndemic theory which states that these conditions act synergistically (correlated, bidirectional, interaction effects). They are part of a syndemic of IPV+substance abuse+HIV. The authors need to clarify that so they do not give the impression that women's alcohol abuse disorders increased the risk for experiencing IPV. Instead, women's substance use disorders co-occur with IPV. IPV victims use substances to numb the effect of traumatic experiences (i.e., IPV) and their perpetrators use their drinking as a way to exert further control over the victims. Alcohol use disorder may increase the vulnerability to become a victim of abuse due to their impaired decision-making. However, the study by Jiwatram and colleagues did not test that as they used "injected drugs" (correlation not significant any way).

In the section "Behavioral risk factors > Psychological well-being" (line 367-369), the following sentence was written: "Psychological well-being: IPV was found to be negatively 367 associated with psychological well being [36];" This sentence must be deleted as it is not supported by the study data. In Ref. 36 (McClintock, H. F., Trego, M. L., & Wang, E. M. (2021). Controlling Behavior and Lifetime Physical, Sexual, and Emotional Violence in sub-Saharan Africa. Journal of interpersonal violence, 36(15-16), 7776–7801), McClintock and colleagues did not examine psychological well-being as risk factor. Instead, they used DHS data (cross-sectional) to assess partner controlling behaviors and their associations with IPV.

Both studies (ref 27 and 36) were cross-sectional. Causality cannot be inferred. This is a major limitation of many studies reviewed.

Perhaps, the reference was 35 and not 36. However, reference 35 was not assessing psychological well-being either as risk but as an outcome instead. John NA, Kapungu C, Sebany M, Tadesse S. Do Gender-Based Pathways Influence Mental Health? Examining the Linkages Between Early Child Marriage, Intimate Partner Violence, and 736 Psychological Well-being among Young Ethiopian Women (18–24 years Old). Youth and Society.

737 2022.

COMMENT 2: This reviewer also agrees with previous review that the section keeps mixing behavioral factors (conduct or life style or addictions) with mental health factors. Behavioral health has more to do with the specific actions people take. It's about how people respond in different life scenarios. Two people who are experiencing depression may react in different ways. Mental health, on the other hand, has more to do with thoughts and feelings (e.g., anxious trait, anxious state; depression; PTSD). Mental health issues can have behavioral manifestations, but those must be delineated in a review like this.

7. PLOS authors have the option to publish the peer review history of their article (what does this mean?). If published, this will include your full peer review and any attached files.

Reviewer #1: No

Reviewer #3: No

---

## [Author Response · Author response to Decision Letter 1]

21 Sep 2023

In the attached 'response to reviewers' file, all comments raised by the editor and reviewers are addressed.

---

## [Editor Report · Decision Letter 2]

10 Oct 2023

Factors and Consequences associated with Intimate Partner Violence against Women in Low-and Middle-Income Countries: A Systematic Review

PONE-D-23-08947R2

Dear Dr. Gunarathne,

We’re pleased to inform you that your manuscript has been judged scientifically suitable for publication and will be formally accepted for publication once it meets all outstanding technical requirements.

Kind regards,

Abraham Salinas-Miranda, MD, PhD

Academic Editor

PLOS ONE

Additional Editor Comments (optional):

The authors have addressed all concerns.
---

## [Editor Report · Acceptance letter]

19 Oct 2023

PONE-D-23-08947R2 

Factors and Consequences associated with Intimate Partner Violence against Women in Low- and Middle-Income Countries: A Systematic Review 

Dear Dr. Gunarathne:

I'm pleased to inform you that your manuscript has been deemed suitable for publication in PLOS ONE. Congratulations! Your manuscript is now with our production department. 

Kind regards, 

on behalf of

Dr. Abraham Salinas-Miranda 

Academic Editor

PLOS ONE